# A Reference Architecture for Enabling Interoperability and Data Sovereignty in the Agricultural Data Space

Rodrigo Falcão *[ID], Raghad Matar [ID], Bernd Rauch [ID], Frank Elberzhager [ID] and Matthias Koch [ID]

Fraunhofer Institute for Experimental Software Engineering IESE, Fraunhofer-Platz 1, 67663 Kaiserslautern, Germany
* Correspondence: rodrigo.falcao@iese.fraunhofer.de

**Abstract:** Agriculture is one of the major sectors of the global economy and also a software-intensive domain. The digital landscape of agriculture is composed of multiple digital ecosystems, which together constitute an agricultural domain ecosystem, also referred to as the "Agricultural Data Space" (ADS). As the domain is so huge, there are several sub-domains and specialized solutions, and each of them poses challenges to interoperability. Additionally, farmers have increasing concerns about data sovereignty. In the context of the research project COGNAC, we elicited architecture drivers for interoperability and data sovereignty in agriculture and designed a reference architecture of a platform that aims to address these qualities in the ADS. In this paper, we present the solution concepts and design decisions that characterize the reference architecture. Early prototypes have been developed and made available to support the validation of the concept.

**Keywords:** digital ecosystem; reference architecture; smart farming; digital farming; architecture drivers; design decisions; solution concepts

## 1. Introduction

A digital transformation is happening and disrupting many major sectors of the economy. The notion of digital transformation includes, among other things, the conception of innovative and digital business models. This affects several traditional industries, such as transportation, banking, and entertainment, which have been impacted by the rise of software companies that approach not simply their products or their processes, but also the fundamentals of their digital business [1]. Agriculture, as a major sector of the economy, is already a software-intensive industry where several players cooperate in huge and complex value chains. At the technical level, the digital transformation in agriculture requires digitally available data from, e.g., the environment, farms, machines, and processes to enable software-supported products and services to work smoothly [2].

However, regarding digital systems, the agricultural domain is rather fragmented [3,4]: There are various systems with various data formats, complying with numerous different standards within its multiple subdomains. Therefore, enabling interoperability in agriculture is challenging [2]. Data are typically distributed across exclusive and isolated data storage systems of suppliers' digital ecosystems. On top of that, there is no or only little semantic interoperability—meaning the ability of applications to exchange data with a shared meaning—which leads to huge efforts in communication and orchestration for delivering complex end-to-end solutions for stakeholders in the agricultural area such as farmers [5].

Furthermore, there is a major concern that is increasingly drawing attention and hindering acceptance of systems by farmers: data sovereignty [4]. This can be observed in several recent studies. In Germany, for example, Bartels et al. [6] verified that farmers feel that they lose sovereignty over their data once they give it to service providers. In Australia, Jakku et al. [7] found trust to be the main concern for farmers when it comes to the design and implementation of smart farming applications. Farmers were highly concerned

with having adequate regulations to protect data privacy and security, transparency, and the ability to know who would benefit from their farm data and how. Furthermore, many expressed their lack of trust in the data ownership assurances expressed by digital agricultural service providers, the transparency regarding which data are being used, for which purpose, and how. Moreover, the farmers' willingness to share their farm data depends on the value and sensitivity of the data. Some of the interviewees expressed concerns about sharing their business secrets (e.g., yield and gross margins), as the misuse of such data can cause huge losses. Yet another study in Australia confirms the problem: Wiseman et al. [8] discussed the need for developing and enforcing legal frameworks that regulate the collection, sharing, and usage of farm data to encourage farmers' adoption of smart farming technologies. In their work, the authors surveyed about 1000 Australian farmers, half of whom were uncomfortable with providing service providers direct access to their data. Furthermore, the majority had little to no confidence that service providers would maintain their data privacy and not share it with third parties. Moreover, many participants communicated that they have little understanding of the terms and conditions of data collection agreements with their service providers and expressed their dissatisfaction with service providers profiting from the data. Another concern of the farmers was the power imbalance in their relationship with smart farming service providers in terms of not having the choice of negotiating the data usage terms.

Despite their concerns regarding data sovereignty, farmers nowadays must rely on multiple digital solutions in order to accomplish their goals in all production steps across the seasons. Mostly, such solutions belong to their own digital ecosystems. Machine manufacturers, for example, usually offer cloud-based systems to channel data collected from the machinery of their respective fleets, which results in distributed data storage if farmers own machines from different manufacturers. In other cases, systems only cover data from their respective business processes and offer no data exchange across farm systems. As a consequence, farmers often face a situation where they have to use different systems with exclusive data vaults, and there is no or only limited connectivity between those systems. From the service providers' perspective, digital services need to integrate different data sources into their respective service environments in order to offer innovative services for farmers. Therefore, not only farmers but all stakeholders continue to yearn for a frictionless yet secure experience.

Although service providers have their own digital ecosystems, together, they form a unique, though diverse, *domain ecosystem*. Kalmar et al. [4] refers to this domain ecosystem as an Agricultural Data Space (ADS) (In this paper, we use both the terms "domain ecosystem" and "ADS" to refer to the digital ecosystem composed of multiple (sub-)digital ecosystems of the agricultural domain.). In light of the complexity and data intensity that characterize the agriculture domain, our research question is:

*How to enable interoperability and data sovereignty in the ADS?*

In this paper, we present eight architecture drivers, six design decisions, and three solution concepts for a reference architecture of a digital platform aimed at addressing these qualities. This work extends our previous paper on the topic [9]: In the previous paper, we had a specific focus on the usage of certain technology (asset administration shells, the so-called "Industry 4.0 digital twins") to address interoperability in agriculture. Back then, one architecture driver and two design decisions were introduced (The architecture driver AD.QS.1 was introduced in the previous paper as "AP.IOP.2". The design decisions DD.1 and DD.6 have their parallels in the design decisions 'DD.2" and "DD.3" of the previous paper, even though they are not exactly the same, for the focus of the previous paper was different.). The reference architecture was created in the context of the research project Cognitive Agriculture (COGNAC) (https://cognitive-agriculture.de (accessed on 17 March 2023).), where we explored applied solutions in field automation, novel sensing, smart devices, and digital data spaces.

The remainder of this paper is organized as follows: Section 2 provides the reader with background information on the concept of digital ecosystems; Section 3 describes the

research method; Section 4 states the architecture drivers; Section 5 introduces the solution concepts of the reference architecture and supporting design decisions; Section 6 presents related work; Section 7 discusses the findings; and Section 8 concludes the paper.

## 2. Background

Digital ecosystems provide a mechanism to bring players together, and their emergence can be observed in numerous domains. Since the term digital ecosystem, combined with the notion of a platform, is used in manifold ways [10], we dedicate this section to defining the terms as a foundation of the reference architecture for enabling interoperability and data sovereignty in the ADS.

A digital ecosystem is a socio-technical system, which means that not only the software is important, but in particular, the players and their relationships with each other [11]. Here, a digital ecosystem is defined by a specific constellation of players, which we will explain in the following, based on the definition of Koch, Krohmer et al. [12].

The construct that connects the players in a digital ecosystem is called a *digital ecosystem service*. At its core, this service is a brokering service that is centered around an asset. An *asset* can be a physical item, such as a used item on eBay, a service, such as an accommodation on Airbnb, or a digital asset, such as music on Spotify. The role that provides the brokering service is called an *asset broker*. This role is not only responsible for the brokering itself but also for providing the means for onboarding the two additional focal roles in a digital ecosystem service: the providers and the consumers of the asset. An *asset provider* places their assets such that they can be found and accessed, after a potential transaction, by *asset consumers*. Beyond these roles, additional players can contribute to an ecosystem service. To distinguish them from the core roles, they are called support providers. *Support providers* can take over any additional responsibility for the provision of an ecosystem service. For example, they can perform fulfillment activities, which are particularly relevant in the case of physical assets; for example, the goods sold at eBay are delivered by parcel delivery services acting as support providers.

The responsibilities of the service broker include the provision of a digital platform. This platform implements the technical aspects of a digital ecosystem in the form of software to enable the ecosystem service, i.e., the brokering of assets between providers and consumers. Figure 1 depicts the main elements of an ecosystem service.

Although we focus on the structural composition of a digital ecosystem and the contributions of different players in their respective roles, the complete depiction of a digital ecosystem requires the additional consideration of aspects related to the business model of a digital ecosystem [13]. Since the subsequent sections of this paper focus on the architectural decisions related to the design of a digital ecosystem and its digital platform, any aspects beyond the technical ones are out of scope.

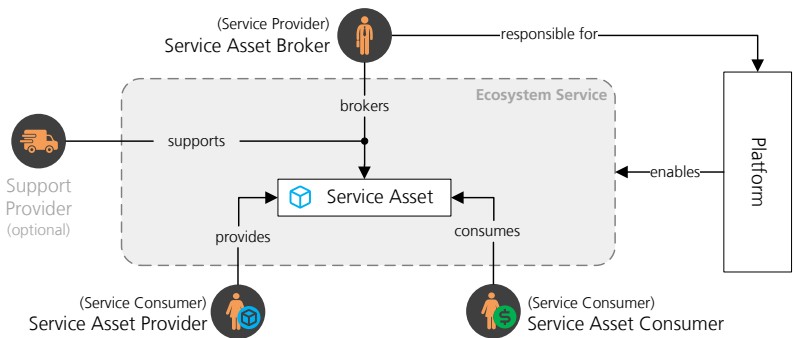

**Figure 1.** Overview of the elements of an ecosystem service [12]. Note that both the Service Asset Provider and the Service Asset Consumer are *service consumers* since they consume the digital ecosystem asset brokering service.

## 3. Method

We used the GQM paradigm [14] to frame our research objective as follows:

*To elaborate a reference architecture for a platform in order to improve interoperability and data sovereignty from the point of view of the data owner in the context of the ADS.*

In order to achieve the goal, we followed the approach for architecture evaluation described by Knodel and Naab [15]. Figure 2 illustrates the method. First, we elicited the architecture drivers related to interoperability and data sovereignty in the context of the project COGNAC. Architecture drivers are a particular type of requirements that focus on what matters most for architecture purposes: business goals, constraints, key functional requirements, and quality requirements [15]. We clustered the elicited architecture drivers into two groups: interoperability and data sovereignty (see Section 4). The drivers, in particular the quality scenarios, emerged not only from our experience in the project COGNAC but also from a feasibility study on Farm Management Information Systems (FMISs) and data management [16]. Furthermore, lessons learned from previous projects that we conducted with both academic and industry partners were considered (see examples in [17]).

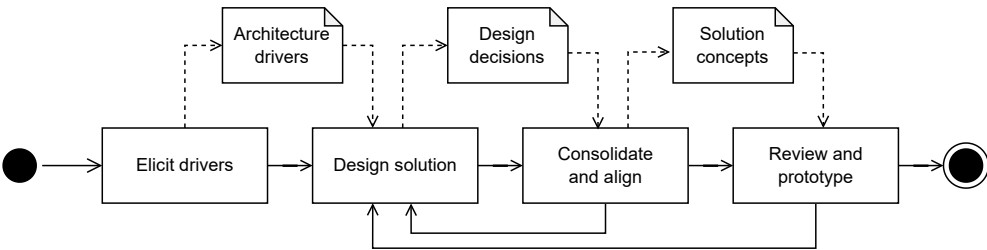

**Figure 2.** Activities performed as part of the research method.

Regarding quality requirements, architecture drivers can be expressed as architecture scenarios. As defined by Rozanski and Woods, an architecture scenario is "a crisp, concise description of a situation that the system is likely to face, along with a definition of the response required of the system" [18]. Architecture scenarios, also referred to as "quality attribute scenarios" [19], ensure that the quality requirement is expressed in a concrete and measurable way. In addition to the quality scenarios, we also described key functional requirements to be provided by the platform because the *raison d'être* of the platform is to enable the quality aspects in the ADS, and this is done through the implementation of concrete functionalities. Thus, there is a mapping between the functionalities implemented by the platform and the domain ecosystem qualities perceived by those who interact with the platform.

Next, we designed the solution by delineating the system context and decomposing functions and data. We used the Architecture Decomposition Framework (ADF) and gave special attention to runtime aspects of the architecture, covering functional, data, and deployment views [20]. Throughout the design process, a series of design decisions were made that support the three solution concepts described in the reference architecture: *field data storage*, *consent and access management*, and *field data exchange*. After that, we consolidated and aligned the design decisions in order to check for inconsistencies (i.e., conflicting design decisions). Finally, we evaluated the reference architecture by means of internal reviews in the project COGNAC and through the implementation of prototypes, which were used by the COGNAC project partners and have been made available as open-source tools (Source code available at https://s.fhg.de/cognac-source-code (accessed on 17 March 2023).) to support the further development of the reference architecture. The design, consolidation, and evaluation activities were fairly iterative.

## 4. Architecture Drivers

Architecture drivers are the architecture-significant requirements of a system to be designed. Therefore, before introducing the reference architecture, it is essential to discuss what drove the solution design. Each architecture driver has an identifier (The acronyms used to identify the architecture drivers have three parts: AD.XX.Y, where "AD" is fixed and stands for "architecture driver", "XX" can be either "FR" (key functional requirements) or "QS" (quality scenario), and "Y" is a sequential number.) and a description. In the case of key functional requirements, the description follows the requirements template proposed in [21], and quality scenarios are expressed in terms of quantified environments, stimuli, and response, as proposed in [15]. In total, there were eight architecture drivers organized into two groups: interoperability and data sovereignty. Constraints and business goals were left out of the analysis, for in our case, these aspects will become more prominent only in future steps once the reference architecture is implemented and business models around its operation must be considered. For example, the technical experience of the team and organizational factors of companies usually play a significant role in constraining technological decisions.

It is also worth noting that there is an intentional mapping between the functional requirements and the quality scenarios. Although they are similar, they differ in two ways. First, while a functional requirement describes what the function is, a quality scenario describes how well the function works in terms of a certain quality. Second, while the functional requirements take the perspective of the farmer, the perspective is shifted to the service provider in all the quality scenarios but AD.QS.4.

### 4.1. Interoperability

In the agriculture domain, interoperability (IOP) relates to software-based systems exchanging data—in particular field data—to perform certain agricultural processes (e.g., fertilization, weed control, etc.), which are usually implemented by service providers.

**AD.FR.1:** The platform should provide farmers with the ability to switch from one service provider to another without any impact on the interoperability of their field data.

**AD.QS.1:** Consider that a service provider operates the service $S_1$, which is already established in the market. In order to provide its service, $S_1$ needs one-time access to read certain field data, which in turn are managed by the farmer through their FMIS. $S_1$ is already capable of getting field data from farmers who use the leading FMISs on the market; however, a new FMIS now enters the market and gains popularity among farmers. One of the early adopters of the new FMIS wants to use $S_1$. Assuming that all accesses have already been granted, $S_1$ should be able to retrieve the required field data from the farmer who uses the new FMIS without the need for any design-time modification.

### 4.2. Data Sovereignty

We define data sovereignty (DS) in the agriculture domain based on three pillars: data portability (the possibility to move data from one system to another), data usage only with consent, and transparency about what happens with the data. In this section, we will describe architecture drivers covering each of these pillars.

#### 4.2.1. Concerning Data Portability

**AD.FR.2:** The platform shall provide the farmer with the ability to host (store) their field data wherever they want; therefore, their field data should not be locked into any particular service provider.

**AD.QS.2:** Consider a service $S_1$ that has access to the size and the crop type of a certain field of a farmer. The field data are stored by the farmer in a self-hosted server $SVR_1$, which is connected to the Internet. The farmer decides to move their field data to a cloud-based server $SVR_2$, which belongs to the infrastructure

of a certain cloud-service provider. The farmer does not have to notify $S_1$ (or its corresponding service provider) about the change. After the change, $S_1$ continues to have access to the farmer's field data as before, without the need to make any design-time modification to the service.

4.2.2. Concerning Data Usage Only with Consent

**AD.FR.3:** The platform shall provide the farmer with the ability to determine *who* has access to *which field data* of *which fields*, *for what purpose*, and *when*.

**AD.QS.3:** Consider that a fertilization recommendation service $S_1$ needs access to the field boundaries, the current nitrogen level, and the crop type of the field in order to be able to deliver a fertilization recommendation. A farmer plans to use the service, but they have not yet granted $S_1$ access to their field data; therefore, $S_1$ cannot access the farmer's field data. At a certain point, the farmer grants $S_1$ one-time access to the required data of one particular field for the purpose of receiving the corresponding fertilization recommendation. As a result, $S_1$ can access the required field data once and is allowed to use it with the sole purpose of providing the desired recommendation.

4.2.3. Concerning Transparency about What Happens with the Data

**AD.FR.4:** The platform shall provide the farmer with the ability to know *who* has accessed *which field data* of *which field*, *for what purpose*, and *when*.

**AD.QS.4:** A farmer has granted many services access to their field data. These grants vary in nature: Some are read-only, others are write-only, yet others are read and write grant accesses. The purpose of the access varies from service to service. At any point in time, the farmer should be able to know which services accessed which of their field data, when, and for which purpose.

**5. The Reference Architecture**

A core asset in the ADS is field data. Service providers need field data to perform their services: Be it a recommendation or the planning for the execution of an agricultural process such as fertilization or harvesting—field data are essential input. What field data are needed may depend on each concrete use case. Examples of field data include field name, boundaries, size, terrain slope, soil type, humidity, nutrient levels, and soil density, to name just a few.

The above examples of field data are clearly associated with the field. However, there is also another type of field data that may not be so obvious at first sight, for these data are not so explicitly bound to the physical field. Examples include the weather in the location where the field is located, work records containing information about previous agricultural processes performed in the field, and crop data such as type of crop, maturity, etc. These examples of *field-related data* are also relevant to support multiple agricultural processes, and therefore we regarded them as part of the core asset of the ADS, i.e., they are also included in our definition of field data.

We highlight two participants in the ADS. First, there are service providers, which play the role of both asset consumers and asset producers. Service providers operate software-based systems and services such as FMISs, fertilization recommendation services, harvesting services, etc., and require field data to work. Conversely, whenever a service is performed (whether it has a direct impact on the field, such as an agricultural process, or not, such as a recommendation), field data are generated. Therefore, in the domain, we have service providers generating and consuming field data from each other. However, service providers do not own the field data they produce or consume. Therefore, the second participant we highlight in the ADS is the farmer, who is the data owner.

The platform described here provides four digital ecosystem services, which can be traced back to the four key functional requirements (see Section 4):

- Facilitation of field data exchange (AD.FR.1);

- Storage of field data (AD.FR.2);
- Field data usage consent management (AD.FR.3); and
- Monitoring of field data access (AD.FR.4).

In the next subsections, we will describe the three solution concepts that compose the platform architecture. The first concerns field data storage, the second, consent and access management, and the third, field data exchange—along with their underlying design decisions. For each design decision, we will discuss pros, cons, assumptions, and trade-offs. Figure 3 shows a conceptual model that depicts the relationships among the architecture drivers (both key functional requirements and quality scenarios), solution concepts, and design decisions (We used the UML notation [22] to depict all diagrams in this paper. In this particular case, for example, we used a class diagram to represent the conceptual model.).

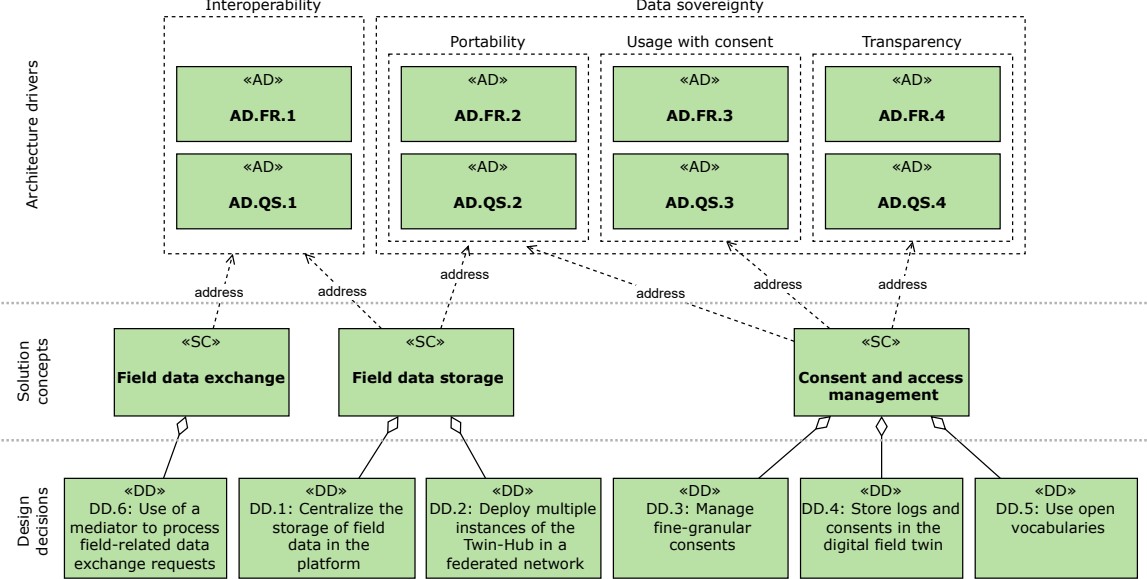

**Figure 3.** Architecture drivers (ADs), solution concepts (SCs), and design decisions (DDs).

### 5.1. Solution Concept for Field Data Storage

The platform centralizes the storage of field data (DD.1 (DD.X is the acronym we used to identify the design decisions, where "X" is a sequential number.)). Although field data can (potentially) be used by several services from different service providers, the platform is the single source of truth for this data. With the data centralized in one place, the farmer is in a better position to manage and monitor the usage of their data (for more on this, see Section 5.2).

We centralize the storage of field data by creating a digital representation of the field in the platform. Therefore, whenever a service provider needs field data, it can rely on the digital representation of the field stored in the platform. The idea of digital representation of real entities is referred to as *digital twins* (DTs). The concept of DTs was coined by Michael Grieves in 2003, referring to virtual representations of physical products with two-way communication between them [23]. In recent years, the idea of DTs in agriculture has been explored as well (e.g., [2,24]). We refer to the digital representations of fields as *digital field twins*—which led us to name the platform "Twin-Hub".

The Twin-Hub has a component called Field Twin Manager, which is responsible for the storage and retrieval of field data. In many cases, field data are static—i.e., they do not change or take too much time to change (e.g., the field boundaries), but there are cases where field data can be quite dynamic (e.g., soil humidity and weather conditions). In such cases, the Field Twin Manager will favor live calls to external services that can provide the latest data and keep a local cache just in case the external service providers are unavailable. Figure 4 shows an initial decomposition of the platform components (further decomposition

will be revealed along with a further description of the solution concepts). The Twin-Hub has a user interface (UI) component, which provides the farmer with the means to manage their field twins, and an API Gateway component, which exposes functions for interacting with the platform.

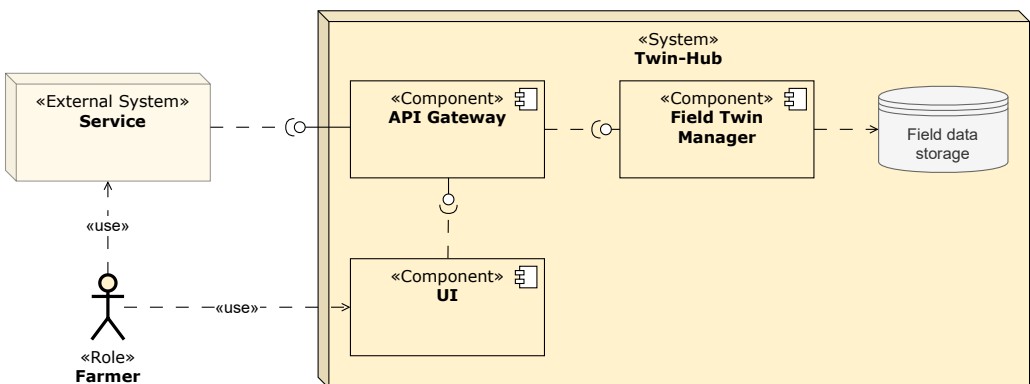

**Figure 4.** Initial functional decomposition of the Twin-Hub.

DD.1. Centralize the storage of field data in the platform:

- **Opportunities/pros:** Digital field twins represent a single source of truth for field data, making data available in a standardized, up-to-date, and non-redundant way. All services can obtain field data from one place.
- **Assumptions:** The digital field twins are hosted in an infrastructure that is reachable through the Internet, and both hosts and digital field twins are identified uniquely.
- **Risks/cons:** In case a digital field twin is unavailable, all services that rely on field data to operate may be impacted.
- **Trade-offs:** Additional dedicated infrastructure and associated costs for implementation, operation, and maintenance are required to deploy digital twins.

Although the field data of a single field is centralized, Twin-Hubs are deployed in multiple instances, creating a federated network of Twin-Hubs (DD.2). Each instance is independent of the others. Therefore, each farmer can decide on which server they want to host their digital field twins. A farmer should keep their fields in a single Twin-Hub, and there should be only one single digital field twin for each real field.

DD.2. Deploy multiple instances of the Twin-Hub in a federated network:

- **Opportunities/pros:** A federated network of Twin-Hubs increases the chances that the platform will penetrate the market in the current domain ecosystem while giving farmers more choices in terms of data protection policies and operation costs, for example.
- **Assumptions:** Digital field twins will have universal unique identifiers. Moreover, there will be parties interested in deploying and operating Twin-Hub instances, ensuring that farmers are offered multiple choices.
- **Risks/cons:** More resources in terms of IT infrastructure are needed in comparison to a centralized approach.
- **Trade-offs:** The evolution of the platform may increase in complexity due to the need to ensure compatibility among different versions of the platform.

Figure 5 illustrates an example deployment diagram. There are three instances of the platform—Twin-Hub 1, Twin-Hub 2, and Twin-Hub 3, and different services (external systems) interacting with the platform instances. Note that nothing prevents the services from communicating with each other directly as they do nowadays. This means that the introduction of the Twin-Hub adds to the ADS without compromising its current working state.

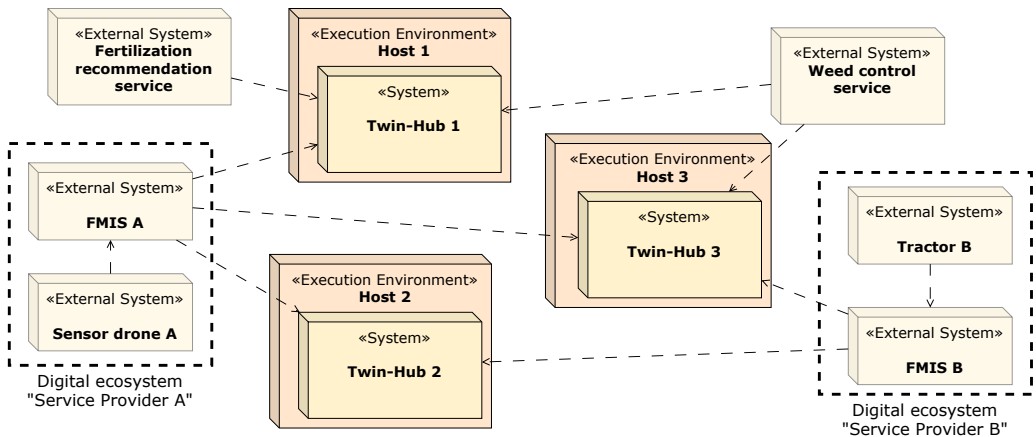

**Figure 5.** Example deployment diagram of a federated network of Twin-Hubs (interfaces are omitted for the sake of simplicity).

*5.2. Solution Concept for Consent and Access Management*

The Twin-Hub implements a Consent Manager component, which is responsible for managing the consents given by farmers to external services interested in their field data. When an external service needs access to field data, the service has to ask the farmer, who is the data owner, for access. For this purpose, the service creates a data access and usage consent request. The two core pieces of information included in the request are the specific field data the service is interested in (e.g., "field boundaries" or "current nitrogen level") and the data usage statement. The former will tell the farmer which fine-granular data they will be granting access to (DD.3) and which type of access (namely: read, create, update, or delete); the latter carries textual information about how the data will be used and for which purpose—information that the farmer will take into consideration to decide whether they will accept or decline the consent request. Since there will be a federated network of Twin-Hubs, the farmer must tell the service the address (URL) of the Twin-Hub that hosts their digital field twins. Then the service is able to send the request to the proper host.

DD.3. Manage fine-granular consents

- **Opportunities/pros:** Farmers' data will be used only as needed—nothing more, nothing less. Service providers can design their solutions to deal with different scenarios—for example, a service could offer the farmer the choice between granting full access to their digital field twins (which would enable more efficiency and/or effectiveness by the service) or minimum access (only the access absolutely required). Fine-granular consent management is also an enabler for transparency about data usage.
- **Assumptions:** Participants in the domain ecosystem will agree upon open vocabularies about field data.
- **Risks/cons:** It may be challenging to reach a community consensus in the domain ecosystem about what "fine-granular" means since different services may have different requirements in terms of data granularity. The implication may be to favor a more fine-grained structuring, which would put an additional burden on service providers that do not need such fine-granular data (even though it can raise more opportunities in the ecosystem for parties that build adapters—for more about this, see Section 7).
- **Trade-offs:** Maintainability of external services (which will need to create fine-granular consent requests).

Farmers manage their consent requests through the UI of their Twin-Hubs. If the farmer has triggered the creation of a consent request, they should notice the request in their Twin-Hubs and can then decide whether to approve or decline it. The acceptance of a consent request creates a *consent*, which grants specific access and usage rights to the service that requested it. The consent contains two levels. The first level is given by the consent request (i.e., the specific field data, the type of access, and the data usage statement). The second level provides information on the fields for which the farmer wants to accept the

consent request, as the farmer may have several fields but may want to grant a particular service access to only a subset of their fields. Figure 6 increments the initial functional decomposition by adding the component Consent Manager. Figure 7 illustrates the process from the creation of a consent request to its acceptance by the farmer in a sequence diagram. The consents given by the farmer to the use of the field data of a certain field are stored in the corresponding digital field twin (DD.4).

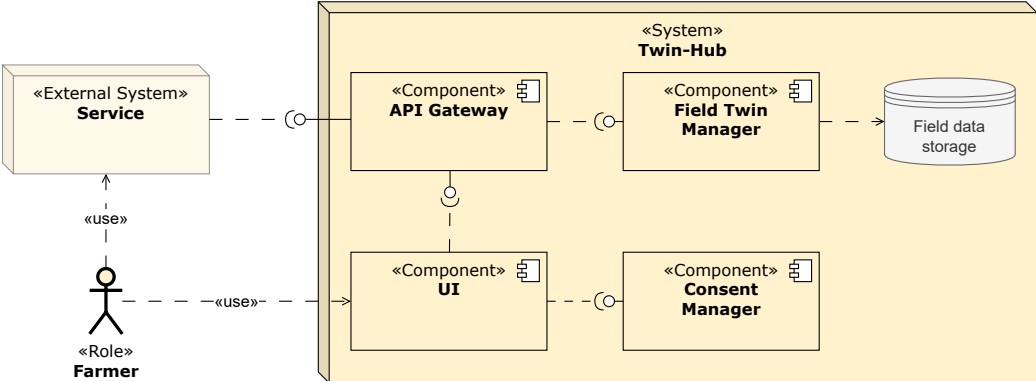

**Figure 6.** Further functional decomposition of the Twin-Hub, now including the component Consent Manager.

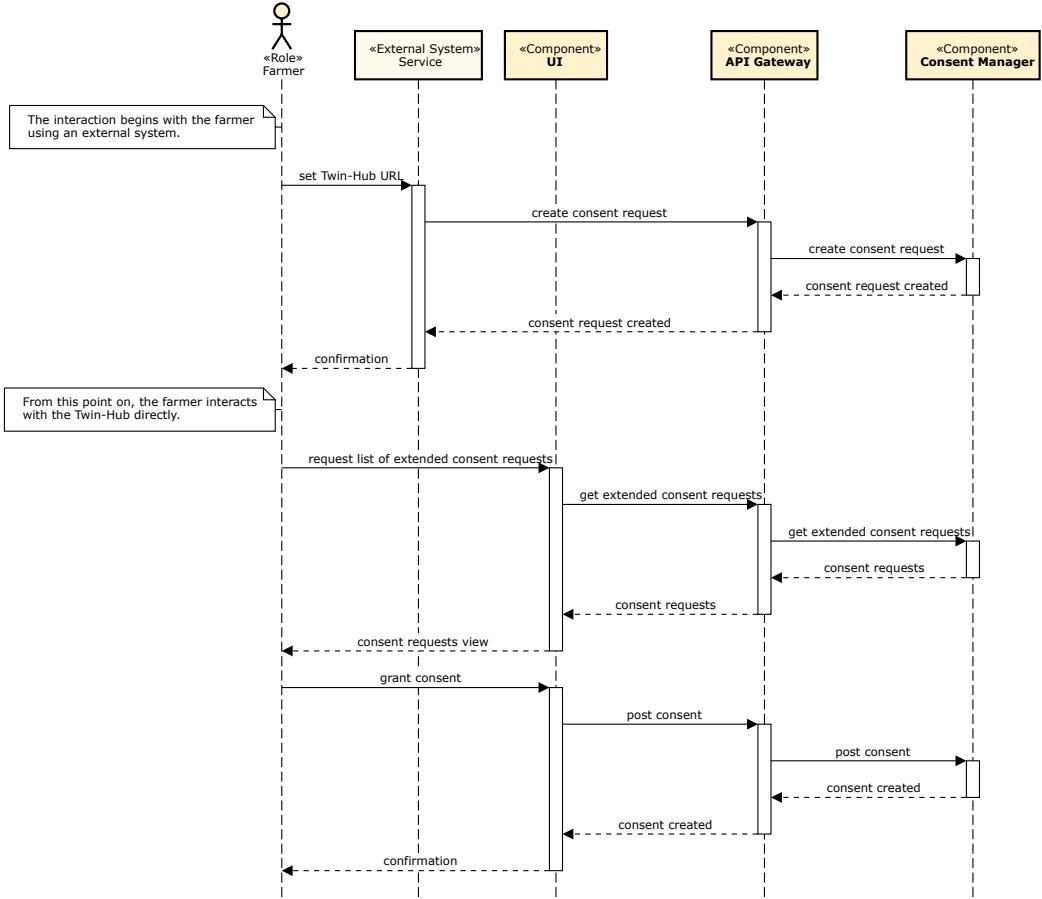

**Figure 7.** Sequence diagram of a consent request. First, the farmer provides the service with the URL of their Twin-Hub instance; next, the service prepares and sends a consent request to the Twin-Hub; then the farmer sees the pending consent request on their Twin-Hub, extended by the list of fields to be included in the consent; finally, the farmer decides to which fields they want to grant the service access and confirms the grant, creating a consent.

Once consent is granted, the corresponding service can interact with the Twin-Hub to create, read, update, and/or delete the specific field data from the specific digital field twins to which they have been granted access. Every time the service reaches the Twin-Hub to access a digital field twin, another Twin-Hub component steps in the Access Manager. It is responsible for checking whether the requester has been granted access to perform the requested operation. For example, if a service $S_1$ requests the reading of the field boundaries of a particular field of a farmer, the Access Manager checks whether there is valid consent that grants access to this operation. Furthermore, anytime the Access Manager allows a service to access a digital field twin, it creates a log entry in the digital field twin containing the timestamp of the operation, the service performing the operation, the user who granted the access, the field, and specific field data, and the operation type. Figure 8 increments the system decomposition by adding the components Access Manager and Logger. Figure 9 illustrates the access control and logging process. Similar to the consents, all field data access logs are stored in the corresponding digital field twin (DD.4).

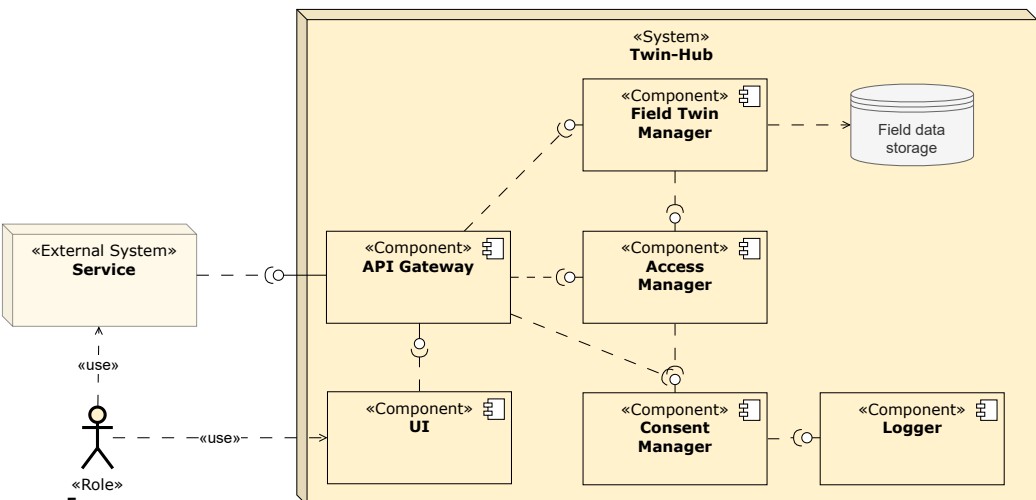

**Figure 8.** Further functional decomposition of the Twin-Hub, now including the components Access Manager and Logger.

DD.4. Store logs and consents in the digital field twin:

- **Opportunities/pros:** Storing logs and consents in the digital field twin improves the portability of the digital field twins. Farmers will be able to export their digital field twins and host them in another Twin-Hub instance of their preference, compromising neither transparency—as information about how the field data have been accessed will come along with the field data—nor consent management—granted field data access will stay valid independent of where the field data are hosted.
- **Assumptions:** When the farmer decides to move their digital field twins from one Twin-Hub to another instance, the former must keep a reference to the latter so that services that request field data on the former instance can be notified about the new instance and adapt their calls accordingly.
- **Risks/cons:** In the case of migration of digital field twins, if the former Twin-Hub instance becomes unavailable (due to failure or decommissioning), services will not be able to find the new host automatically. In such scenarios, the farmer will need to reconfigure their services manually.
- **Trade-offs:** Digital field twins will be inflated with data that do not belong to the real entities (the fields).

Different services may represent field data in different ways. Therefore, when a service creates a consent request, it can refer to the specific field data it is interested in, not necessarily as they are represented in the digital field twin, but through shared terms of an open vocabulary, which must be reachable by all participants in the ADS, including the

Twin-Hub instances (DD.5). Thus, when, for example, a service $S_1$ requests the field area in m$^2$, the request should refer to a certain data item $d$, which in turn points to an entry in an open vocabulary (e.g., AgroVoc (https://agrovoc.fao.org/browse/agrovoc/en/ (accessed on 17 March 2023).) and AgroPortal (http://agroportal.lirmm.fr/ (accessed on 17 March 2023).)) that explains the semantics of the data.

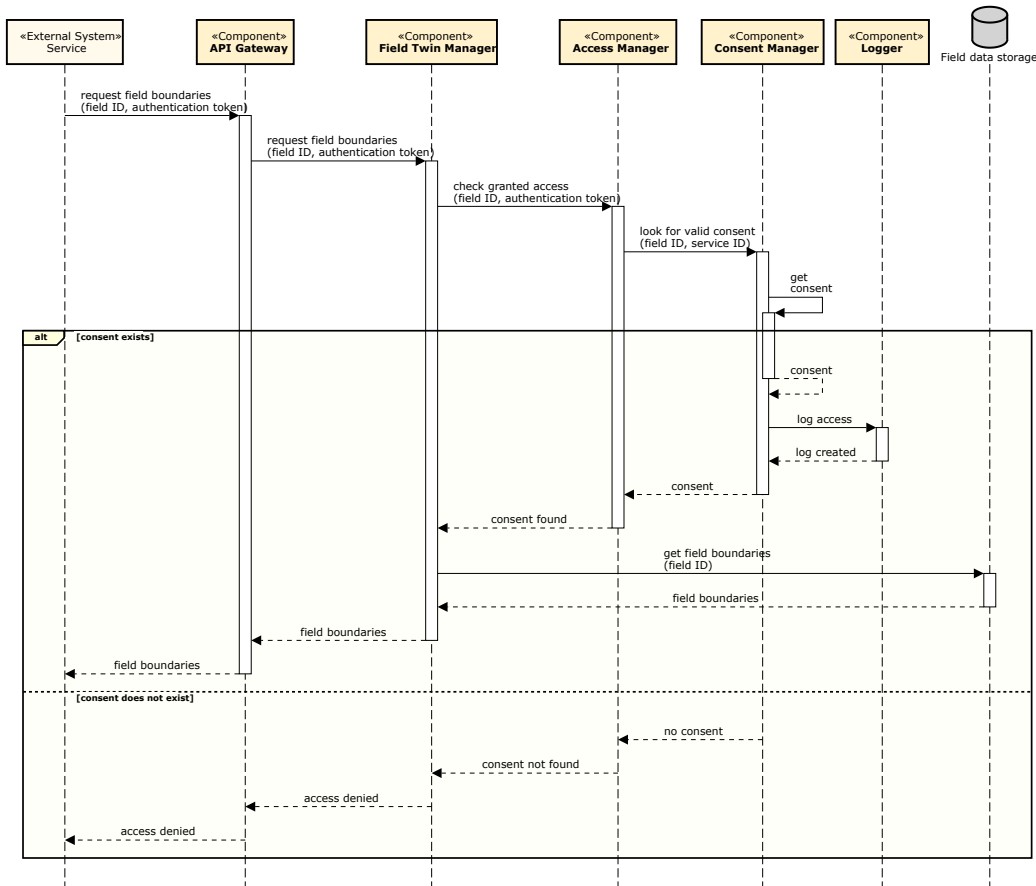

**Figure 9.** Sequence diagram of the logging process. First, a service requests the reading of the field boundaries of a certain field. The request reaches the Field Twin Manager through the Twin-Hub API. Next, the Field Twin Manager asks the Access Manager whether the access has been granted to the requester. Then the Access Manager checks with the Consent Manager whether there is valid consent for the request. After that, if there is valid consent, the Access Manager logs the access. Finally, the Field Twin Manager retrieves the field boundaries from the corresponding digital field twin and returns this data to the requester.

DD.5. Use open vocabularies:

- **Opportunities/pros:** Open vocabularies enable a shared understanding of the meaning of the data. Once the field data has been semantically annotated, field data operations such as consent management and field data exchange can be based on the semantics of the data.
- **Assumptions:** All field data being exchanged through and stored in digital field twins must be semantically annotated.
- **Risks/cons:** In some cases, it may be challenging to reconcile the fine granularity of raw data and the high-level nature of semantics.
- **Trade-offs:** Services must adapt to working semantically, which is not the state of the practice in terms of interoperability in the domain. This means that the ecosystem onboarding effort may increase.

### 5.3. Solution Concept for Field Data Exchange

The Twin-Hub API provides standardized access to field data, meaning that service providers need to know the interface in advance and must adapt their services to read and write field data accordingly. This field data exchange strategy, which is synchronous, is illustrated in Figure 9.

However, it may be that service providers do not know in advance the interface of the digital field twin exposed through the Twin-Hub. For this reason, we added a second field data exchange strategy to the solution: When a service wants to read or write field data to the digital field twin, they can request a data exchange through a generic operation, instead of posting/getting data directly to/from a server (DD.6). A data exchange request does not post or obtain field data, but requests the platform to do so. This operation takes as input parameters about the operation type (read, create, update, or delete field data), the list of specific field data involved in the data exchange (e.g., field area, work record, etc.), and the filtering information to determine which records will be impacted by the operation (e.g., a READ operation could ask for the field boundaries of certain specific fields, or for work records of a certain type).

The data exchange strategy is implemented by a Twin-Hub component called Data Exchange Manager. The component combines characteristics of the architectural design patterns Mediator and Command [25]: It facilitates field data exchanges between the digital field twins and the external services by handling data exchange requests. When a service requests a field data exchange, the service must expose an API to support the operation. For example, if the service wants to read field data from the digital field twin, it should expose an operation in its API so that the Data Exchange Manager can call it to send the required data to the service; conversely, if the service wants to store data in the digital field twin, the service must expose an operation in its API to make the data accessible for the Data Exchange Manager, which in turn will store the data in the digital field twin afterward. Internally, the Data Exchange Manager component is composed of two sub-components: the Data Exchange Register (which receives and records data exchange requests) and the Data Exchange Processor (which asynchronously processes the data exchange request). Figure 10 shows the final functional decomposition of the Twin-Hub (including the component Data Exchange Manager), and Figure 11 illustrates the strategy in a sequence diagram.

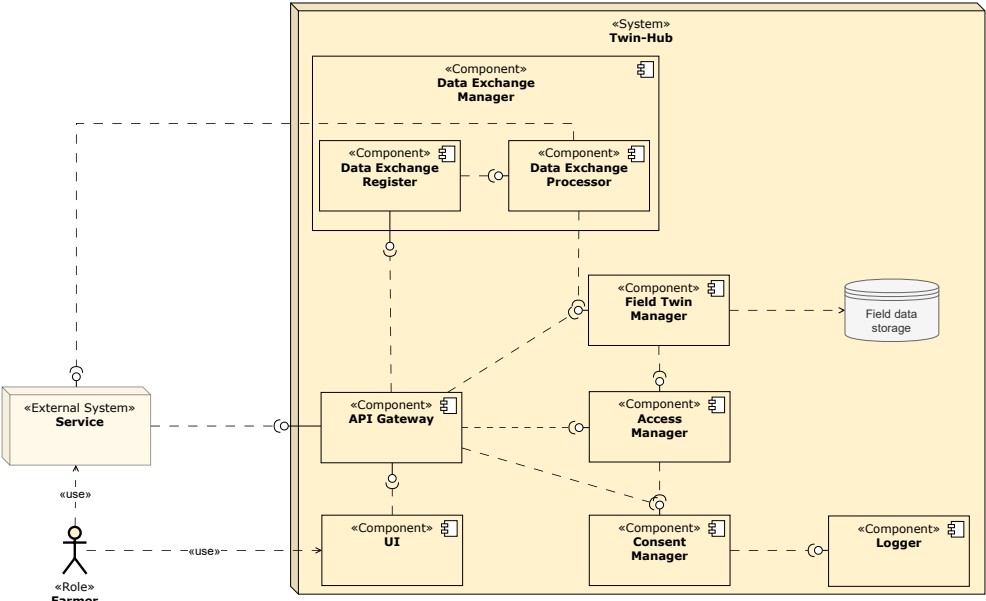

**Figure 10.** Final functional decomposition of the Twin-Hub.

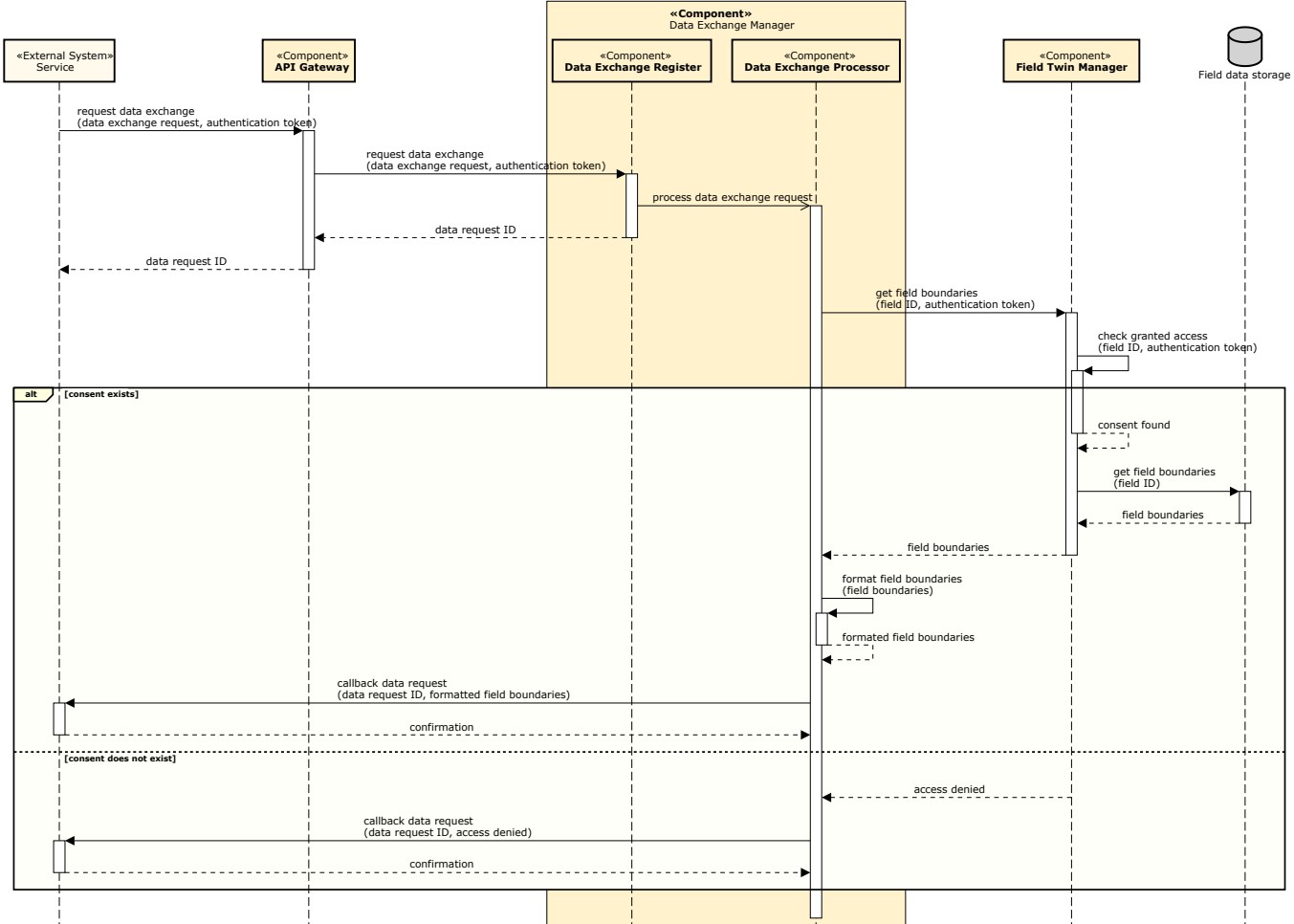

**Figure 11.** Example of the data exchange strategy for a READ operation. First, a service requests the reading of the field boundaries of a certain field. Next, the request reaches the Twin-Hub through its API and is forwarded to the Data Exchange Manager, which stores the data request. Then the Data Exchange Manager asks the Field Twin Manager for the data, which in turn retrieves the data from the storage. Finally, the Data Exchange Manager formats the field data and calls the requester back with a request to send the requested data.

DD.6. Use of a mediator to process field-related data exchange requests:

- **Opportunities/pros:** Systems that are interested in field data do not have to comply with any specific interface provided by the digital field twin. New field data can be incorporated into the twin through generic commands.
- **Assumptions:** All participants agree on the usage of an open shared vocabulary providing the data with semantics.
- **Risks/cons:** There is a risk that the usage of generic constructs will make the usage of the mediator complex, depending on the data involved.
- **Trade-offs:** The complexity of the interaction between systems is replaced by the complexity of the mediator itself (as foreseen in [25]). Furthermore, the overhead caused by data transformations and the need for multiple calls to transfer complex data may impact performance.

Prototype

To build confidence in the design of the platform, we developed a prototype based on the reference architecture. This not only adds to its evaluation but is also a step towards the development of a reference implementation for the reference architecture. The pro-

totype has been continuously developed. At the time of this submission, the following functionality has been developed:

**Centralized field storage in digital field twins.** We used a reference implementation for asset administration shells (as digital twins are referred to in Industry 4.0) to implement the digital field twins. Access to the digital field twins takes place through an API Gateway, which provides standardized operations to access field data. In Figure 12, we present screenshots of two different services accessing the field boundaries of a certain digital field twin hosted in an instance of the Twin-Hub prototype. These services were developed by different teams in the project COGNAC.

**Twin-Hub UI, Consent Management, and Access Management.** The prototype includes a UI through which the farmer can currently manage their digital field twins; accept (or decline) consent requests; visualize (and revoke) granted consents; and visualize the access log. Figure 13 shows excerpts of a service and the Twin-Hub UI to illustrate the user journey from the creation of a consent request to its acceptance and access visualization.

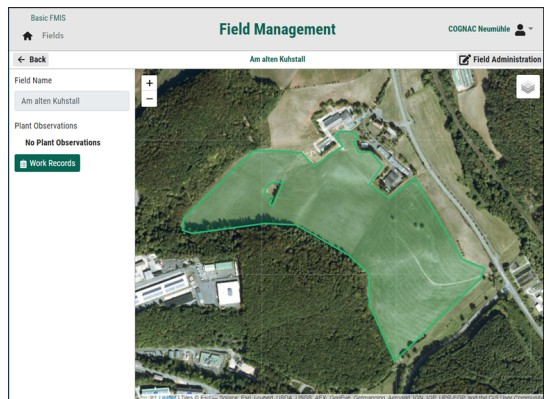
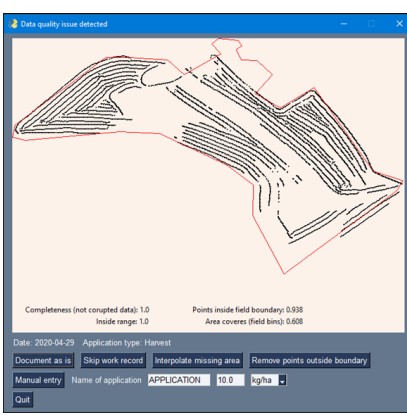

(**a**) Screenshot of *Basic FMIS*.      (**b**) Screenshot of *N-Doc*.

**Figure 12.** Two different services accessing the same digital field twin.

**Connect to my Field Twins**

Please provide the address of your Digital Field Twin Hub Instance:

> Twin Hub URL

Please provide a valid Twin Hub URL.

To provide our services, we need to access your Digital Field Twins. This requires your explicit consent. Please choose below what level of access you would like to provide:

◉ **Recommended:** Full-Access
To offer all features, we need full access to your field twins.

○ **Medium:** Read access to fields, their boundaries, stored work-records and plant-observations. Limited access permissions will result in limited features provided by our service.

○ **Minimum:** Read access to fields and their boundaries. This is the minimum set of permissions required to provide our service with severely limited features.

Once the consent request is accepted at the Twin Hub, Basic-FMIS will be able to access the field twins.

**Data Usage Statement:**

Basic-FMIS is an FMIS developed by Fraunhofer IESE. Basic-FMIS provides you with the means to manage your fields and associated field related data. To do so, Basic-FMIS requires access to your Digital Field Twins located at the Digital Field Twin Hub of your choice.

We need read access your Digital Field Twins in order to display their data to you. We need write access your Digital Field Twins in order to populate the data and changes that you contribute using Basic-FMIS. In order to manage the Digital Field Twin entities (i.e., creating new Field Twins, changing the name of a field,

Close   Confirm

(**a**) Creation of a consent request in *Basic FMIS*.

**Figure 13.** *Cont.*

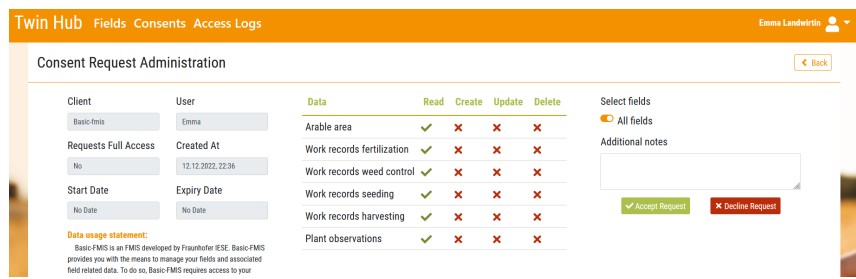

(**b**) Visualization of consent request in the *Twin-Hub UI*.

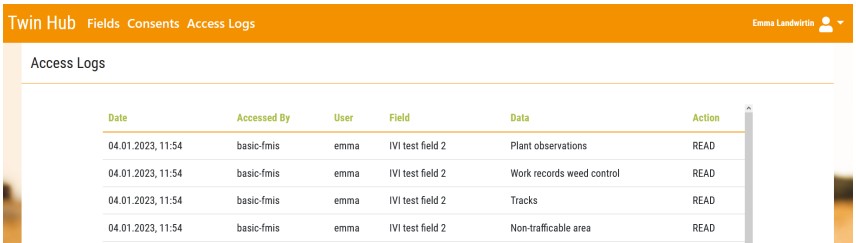

(**c**) Visualization of data access in the *Twin-Hub UI*.

**Figure 13.** Screenshots of the users' journey through consent management.

## 6. Related Work

When we searched for existing reference architectures of digital ecosystems in the agricultural domain in the literature, we found that it is a narrow field whose exploration is in its early stages. To search for existing related work, we used the search engine Scopus (https://scopus.com (accessed on 17 March 2023).), which indexes the results of both IEEE Xplore (https://ieeexplore.ieee.org/ (accessed on 17 March 2023).) and ACM Digital Library (https://dl.acm.org/ (accessed on 17 March 2023).), two of the most popular search engines in software engineering (but it is not limited to these two). Table 1 shows the aspects we covered and the terms we used for each of them. The search string ensured that at least one term of each aspect would be included. Note that we combined terms concerning "reference architecture" and "digital ecosystems" because separating led to nearly no results.

**Table 1.** Aspects covered by the search string, along with the chosen terms applied.

| Aspect | Terms |
|---|---|
| reference architecture and digital ecosystem | "reference architecture" OR "digital eco system" OR "digital ecosystem" OR "data space" |
| agriculture | "agriculture" OR "digital farming" OR "smart farming" |

The search engine returned 84 papers. We analyzed the results in three rounds in order to remove any false positives. First, we evaluated the papers based on the title and keywords and excluded irrelevant ones, i.e., those that are clearly not related to the object of investigation. Then we went through the abstract, introduction, and conclusion and filtered out other results. Finally, we filtered the papers based on the content. We excluded papers that did not contain the term "ecosystem" and those that limited their scope to only specific types of systems in agriculture (e.g., studies that introduced a reference architecture for farm management information systems). As a result of the third round, we found three papers that introduced generic concept solutions for digital agricultural ecosystems. In the following, we will briefly summarize these studies.

Roussaki et al. [26] introduced the architecture of DEMETER, an approach for building an interoperable space for smart agriculture to enable data sharing across heterogeneous

systems and data resources while placing control in the hands of the farmers regarding the usage of their data and the resulting knowledge. The reference architecture of DEMETER is based on four architectural concepts: building a Stakeholders Open Collaboration Space (SOCS) system for the farmers to express their needs in a central framework and for multiple stakeholders (e.g., agricultural advisors, technical services providers) to collaborate on defining appropriate solutions; building an Agricultural Interoperability Space (AIS) where technology providers can cooperate to co-create interoperable solutions for the identified challenges in the domain; the DEMETER Enabler Hub (DEH), which provides a semantically consistent description of registered entities (applications, services, platforms, and hardware) and enables secure access to these resources via identification verification; and a user-friendly dashboard as an entry point for accessing the resources of the ecosystem by the different stakeholders and enabling account, access, and data management. DEMETER enables interoperability at the data level. To achieve this, it implements the Agricultural Information Model (AIM), a semantic data model that extends existing agri-tech ontologies and vocabularies to support agricultural data management and translation. The authors report the results of the first round of instantiating the DEMETER architecture in two pilots in the arable crops sector.

In another study, Budaev et al. [27] propose a smart farming solution for precise agriculture as a knowledge-based digital ecosystem (system of systems) in which each smart service is an autonomous multi-agent AI system. The proposed service-oriented architecture is based on peer-to-peer service interactions and consists of multiple agents of services: agents of fields, monitoring, machines, pesticides, fertilizers, and finance. New services can register themselves to join the ecosystem. The agents of services can react to events and negotiate with other agents to resolve conflicts and reach a consensus on a coordinated decision. The main part of the proposed solution is the precise farmers' knowledge base (e.g., soil types, wheat types, classes of machines, plant diseases, insects, pesticides, fertilizers). The knowledge base consists of two components: an "Agricultural Domain Ontology" component describing the concepts and relations in the domain, and the "Enterprise Ontological Model" component, an instantiation that reflects the status of each enterprise joining the ecosystem. Thus, when a new farmer joins the ecosystem to consume services, they create an instance for their enterprise and provide information about the history and current status of its objects, e.g., fields and machines. To validate the concept, the authors designed and tested a prototype for more than 3000 ha of wheat fields.

Kruize et al. [28] introduced an ontology for farm software ecosystems and proposed a reference architecture for interoperable farm software ecosystems. In their work, the authors focus on arable farm enterprises. The proposed reference architecture facilitates the coherent integration of ICT components (e.g., sensors, implement assemblies, software applications, etc.) from different vendors and seamless data exchange between these components. The architecture contains a modular open platform that facilitates collaboration between the actors in the ecosystem. The platform has a "security, privacy, and trust" module to manage communications in a secure manner using authentication and authorization methods. The platform also has another module (the "system and data integration" module) that provides APIs and data mediation mechanisms for smooth data exchange between application components on the platform. Another component in the architecture is a neutral, open software enterprise that provides the infrastructure to enable this ecosystem and manages the relationship between the actors and the platform. The researchers identified multiple actor roles in the farm software ecosystem: software vendors, agricultural service providers, infrastructure providers, and the end-users or the customers who use the services and the software components of the ecosystem, e.g., farmers, contractors, and agronomists. The actors can provide or use the configurable ICT components and the business services offered on the platform. Another role in the ecosystem is the orchestrator that runs the platform. The concept of a farm software ecosystem that facilitates the configuration of customized systems utilizing standardized components from different vendors can prevent the big players from dominating the market and eliminate the vendor lock-in

problem for farmers. Furthermore, the authors recommend using data standards to enable the portability of the data between different farm software ecosystems and thus reduce the possibility of ecosystem lock-in. To evaluate the reference architecture, the authors verified its design against a set of requirements for farm software ecosystems, which they identified based on their literature analysis and the empirical results of two projects related to the agricultural domain. The analysis showed that the requirements were addressed by the architecture. Furthermore, the authors carried out a conceptual validation by using the proposed architecture to map two existing farm software ecosystems (Crop-R and AgroSense) based on semi-structured interviews conducted with the CTOs of the two ecosystems. Many components and subcomponents of the architecture were present in the two analyzed ecosystems. The mapping proved to be helpful for analyzing existing ecosystems and identifying their similarities and differences. However, the evaluation did not include assessing whether the proposed reference architecture is useful for evaluating, designing, and implementing new farm software ecosystems.

## 7. Discussion

### 7.1. Opportunities and Challenges

Architecting digital ecosystems is a multidisciplinary task that comprises, among other things, organizational and technical aspects. Although we have focused on the technical aspects of a reference architecture for a platform in the ADS, making it practical adds organizational challenges to the initiative. From a technical perspective, this paper argues for a decentralized architecture to improve two quality aspects of the ADS—interoperability and data sovereignty. In this domain, there are already multiple players dealing with slices of various subdomains and several supporting and competing platforms.

From a purely technical perspective, a centralized approach could be proposed. Considering, for example, the software qualities listed in ISO 25010 [29], the advantages of a centralized platform include streamlined modifiability, operability, confidentiality, and integrity. Furthermore, such an approach also lowers the requirements for installability and adaptability. However, organizational aspects challenge this proposal. First, the agricultural domain is huge and scattered, meaning that a centralized platform for managing all processes in smart farming is just not practical. The next issue is competition: Major players have developed their own (sub-)digital ecosystems and aim at having farmers' data more and more concentrated within their realms [4]. The question is: Who would be in a position to ultimately operate such a centralized platform for the benefit of all participants—including both farmers and multiple (and competing) service providers?

For this reason, we see better chances for the flourishing of an enabling platform in the ADS if it is going to be an open platform. Parker et al. [1] refer to platform openness in terms of manager and sponsor participation, developer participation, and user participation. Therefore, we see a prominent role of research in the further improvement of not only the design but also the development of a reference *implementation* of the Twin-Hub, which must be a cornerstone for an open domain ecosystem. We believe that it should be possible for many parties—including farmers themselves—to deploy and operate their own Twin-Hub instances. This is particularly important for data sovereignty reasons. On the one hand, some companies are currently seizing upon the existing interoperability issue to offer data routing platforms that work as adapters, converting data from one service provider's format to that of others. On the other hand, assuming that their offers are broad enough to cover a large variety of formats from several providers in the domain and fast enough to adapt to changes (e.g., the appearance of a new participant such as a novel FMIS provider), this may only tackle the interoperability aspect discussed in Section 4—in fact, this is the state of the practice and does not suffice to support thorough connectivity across the domain [4]. The issue of data sovereignty would continue to persist, as can be seen in practice today.

The development of a reference implementation of the Twin-Hub as an open platform will also unlock many business opportunities in the domain, as it has the potential to enable

multiple marketplaces around field data—and opportunities might be available not only for big players: Competition will also be open for startups as well as small and medium-sized companies. We envision both service-provider-oriented marketplaces and farmer-oriented marketplaces. Each marketplace could work as a third-party service connected to multiple Twin-Hubs. For example, for service providers, there might be a marketplace for raw field data, which could be of value for companies who train machine learning models; another marketplace might provide these machine learning models (e.g., weed recognition models) to service providers offering weed control services to farmers; yet another marketplace might offer data adapters for transforming data across different formats. For farmers, apart from a marketplace for agricultural services, there might also be potential marketplaces for "service templates" or "service bundles", i.e., abstractions for service pipelines in which the farmer might be interested. This means that farmers could either contract multiple service providers individually, on their own or use these abstractions to contract composed offers providing end-to-end services.

### 7.2. Comparison with Related Work

Our reference architecture is unlike related work in many ways, three of which deserve particular attention: (1) the type of ecosystem at which the reference architecture is aimed; (2) the type of enabling platform described in the reference architecture (and the services provided by the platform); (3) how each reference architecture addresses the architecture drivers presented in Section 4. Table 2 summarizes the differences.

With respect to the ecosystem type, most reference architectures focus on the domain ecosystem, except for Kruize et al. [28]—in this particular case, the authors' main focus is rather on individual digital ecosystems (referred to as "Farm Software Ecosystems"). In fact, they affirm that "[In a scenario with competing ecosystems and their platforms,] a data broker could be useful to exchange (and maybe store) data used by both platforms".

The type of platform and how the architecture drivers (interoperability and data sovereignty) are addressed are intertwined. Unlike our work, all other works favor a centralized platform. A centralized platform has its advantages, as discussed in Section 7.1, but the pros and cons must always be balanced according to what is perceived to be the most appropriate solution to address a set of drivers. Although the other works present solution concepts, they do not clearly present rationales for the underlying design decisions, so we can only speculate about their reasons for choosing a centralized platform. One possibility may be the fact that, although they did focus on interoperability, they did not take the farmer's perspective, which is key for the data sovereignty drivers. Indeed, in their reference architectures, the platform is centralized, but the agricultural data are not; instead, this data remains distributed across the different solutions of the different service providers. In their reference architectures, the role of the platform is rather to make the different services from different service providers discoverable. Furthermore, they also deal with adapting, converting, and/or integrating data from different formats. Conversely, our reference architecture contributes a solution where the platform is decentralized, but it *centralizes* the data (in our case, field data) in its instances. This was important for us in order to address the three aspects of data sovereignty, which were neither properly discussed nor addressed by the other reference architectures. It is worth noting that the reference architecture presented by Roussaki et al. [26] discusses components to enable semantic interoperability in much more detail. Actually, in the paper, the authors do not provide much detail about the reference architecture but rather a high-level view of it. However, the paper contains a reference to a project report [30] where the authors did present their reference architecture in more detail than ours. These can be used to implement the semantic aspect of interoperability of our reference architecture.

**Table 2.** Comparison between related work and our reference architecture.

| Work | Ecosystem Type | Type of Enabling Platform | Interoperability | Data Portability | Data Usage Only with Consent | Transparency |
|---|---|---|---|---|---|---|
| Roussaki et al. [26] | Domain ecosystem | Centralized | Distributed data storage, standardized access (common information model, but extensible) | Indirectly discussed, and from the point of view of service providers: participating services (external to the platform) must be motivated to share their data | Partial: data usage policies are managed by the platform; access control refers to access to the DEMETER-enabled entities, not the data itself. | Not discussed |
| Budaev et al. [27] | Domain ecosystem | Centralized | Not discussed | Not discussed | Not discussed | Not discussed |
| Kruize et al. [28] | Digital ecosystem | Centralized | Distributed data storage, standardization (accepts multiple standards) | Indirectly discussed, and from the point of view of service providers: participants must adhere to standards | Not discussed | Not discussed |
| This work | Domain ecosystem | Distributed (multi-instance) | Two-fold: centralized data storage with standardized access, and generic data exchange mechanism | The data owner—i.e., the farmer—has the autonomy to host their digital field twins wherever they want, and move their digital field twins from one place to another | Partial: solution consists of granular consent management (currently addresses only data access control; data usage control is future work) | Monitoring mechanism included |

*7.3. Limitations*

The reference architecture presented in this paper contributes to paving the road that leads to the realization of a better ADS; however, it has its own limitations. One limitation is given by the clear focus placed on interoperability and data sovereignty—although important, these are not the only quality attributes that need to be taken into consideration. For example, we point out scalability as a crucial quality to be addressed, as the volume of data (in particular raw data) collected in field operations can be extremely high depending on the nature of the operation and the sensing technology involved. Moreover, with respect to the scope of the data, so far, we have concentrated our efforts on field data, which plays an important role in agriculture, in particular for arable farming. However, we recognize that there is more than field data being exchanged in the domain. Although our focus on field data fits better with arable farming, we believe that the concept of digital twins can also be applied to other entities in the domain, such as plants or livestock.

Another limitation present in the reference architecture refers to the technical enforcement of data usage control. From a technical point of view, the architecture provides data *access* control, not data *usage* control. As technical enforcement of data usage policies becomes hard once the data leaves the platform, organizational measures (e.g., terms of use, legal contracts, etc.) must be put in place. Technical enforcement of data usage control can be improved by applying the strategy of moving algorithms to data. The application of this strategy to our design would mean that data would not leave the Twin-Hub, but instead, service providers' algorithms would move to the Twin-Hub, run on the farmer's data, and provide service providers only with the results (when applicable). Moving algorithms to data is a known way to protect data [31,32]. To make this possible, the platform could have a dedicated component to manage the execution of external algorithms, which has not been designed in the current reference architecture. Alternatively, IDS connectors—a software component for handling, among other things, the management and enforcement of data usage control [33]—could be used in front of the parties to mediate data and algorithm exchange.

The evaluation of the reference architecture is limited by the internal reviews and the current status of the prototype. It is worth noting that in the current implementation, there is no Data Exchange Manager (see Section 5.3), which is a key component for enabling semantic interoperability of field data. We plan to follow an incremental strategy to build the component, starting with basic field data such as field boundaries, name, and area (which are useful in many use cases). Furthermore, it will be necessary to decide on the initial vocabulary to support semantic interoperability (most likely using or extending existing ones—see DD.4 in Section 5.2).

In this work, we focused on the technical aspects of the platform, but we do recognize its limitations with respect to organizational aspects that must be taken into consideration. We point out the need for dedicated research on the development of business models for the Twin-Hub as well as for the services around it: The roles of other participants in the ADS (e.g., the platform operator, public authorities, research organizations, customers, etc.) must be made clear, money flow among parties must be devised, and a platform governance model must be created.

## 8. Conclusions

In this paper, we presented the reference architecture of the Twin-Hub, a multi-instance platform aimed at addressing interoperability and data sovereignty in the agricultural domain ecosystem. The solution features centralized field data storage in digital field twins, meaning that for each real arable field, there will be a unique digital field twin hosted by one instance of the platform. Data access is controlled through a fine-granular consent management mechanism that uses a shared vocabulary; i.e., services ask for the specific field data they need and receive access to it accordingly. All data access operations are logged, and both logs and consents are stored along with the field data in the digital field

twin. This means that if a farmer decides to switch from one Twin-Hub instance to another, transparency about who has access to their data is preserved; likewise, data access consents are also portable.

As the architecture of the Twin-Hub will become increasingly improved towards supporting a reference implementation, we are aware that future research directions should include a broader assessment of the architecture maturity that goes beyond prototyping. We have already worked on the development of a reference implementation for the Twin-Hub, whose progress has been made available as open-source software. We regard the provision of a fully functional reference implementation of the Twin-Hub as an important contribution that research can make to improve interoperability and data sovereignty in the agricultural domain. We highlight the importance of further designing and implementing the Data Exchange Manager, which is expected to be the most challenging component to realize.

Beyond the development of the reference implementation, one of the future directions we see is providing the capability to host digital twins of entities other than fields. Furthermore, it is necessary to investigate how to address other quality attributes (e.g., scalability) and also to investigate business models for operating Twin-Hubs. As we step into the business models topic, the use of another language to express the architecture may also be helpful to support future directions. In this respect, alternatives include Archi-Mate (https://www.opengroup.org/archimate-forum/archimate-overview (accessed on 17 March 2023).) for its in-built support for describing the business and strategic aspects.

**Author Contributions:** Conceptualization, R.F. and B.R.; Investigation, R.F. and R.M.; Project administration, B.R.; Writing—original draft, R.F., R.M., B.R. and M.K.; Writing—review & editing, B.R. and F.E. All authors have read and agreed to the published version of the manuscript.

**Funding:** This research was funded by the Fraunhofer-Gesellschaft in the context of the lighthouse project Cognitive Agriculture (COGNAC).

**Data Availability Statement:** Not applicable.

**Acknowledgments:** We are grateful to Sonnhild Namingha from Fraunhofer IESE for proofreading this article.

**Conflicts of Interest:** The authors declare no conflict of interest.

## Abbreviations

The following abbreviations are used in this manuscript:

| | |
|---|---|
| AI | Artificial intelligence |
| AD | Architecture driver |
| ADS | Agricultural Data Space |
| COGNAC | Research project "Cognitive Agriculture" |
| DD | Design decision |
| DT | Digital twin |
| DS | Data sovereignty |
| FMIS | Farm Management Information System |
| FR | Functional requirements |
| GQM | Goal, Question, Metric |
| ICT | Information and communication technologies |
| ID | Identification |
| IDS | International Data Spaces |
| IOP | Interoperability |
| QS | Quality scenario |
| SC | Solution concept |
| UI | User interface |

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
