# Peer review of "A Reference Architecture for Enabling Interoperability and Data Sovereignty in the Agricultural Data Space"

_information, doi:10.3390/info14030197_

Round 1
Reviewer 1 Report
In this manuscript, the authors propose a reference architecture to enable interoperability and data sovereignty in agricultural data spaces. In particular, the authors identified architecture drivers indicating the requirements of the reference architecture. Then, they discussed how the proposed architecture can address them. Finally, they validated the proposed architecture in the context of the COGNAC project, both by interviewing the participants and by creating a software prototype.
The manuscript is well written and organized, and is easy to read and follow. However, it presents a few issues that should be better addressed. Firstly, the authors should better clarify the differences with respect to their previous publication [9]. In the current version, they mention that this work is an extension of [9], but they don't specify which aspects have been further developed in this manuscript.
Similarly, when surveying the state of the art, the authors summarize the related work, without clearly clarifying their shortcomings and to which extent their proposed reference architecture can address them. To make this comparison clearer, I would suggest the authors to move the related work section after presenting their reference architecture (Section 6), commenting to which extent such existing work addresses the drivers identified. To this aim, a table comparing the state of the art and the proposed architecture would help.
Finally, the diagrams being used to represent the software architecture and the relations between drivers, requirements and architectural decisions do not follow a specific modeling language. It would be helpful if they would instead adhere to some specifications, such as UML or ArchiMate.
Reviewer 2 Report
Dear authors,
Thank you for submitting your paper on the reference architecture for interoperability and data sovereignty in agriculture. I have carefully reviewed your work and have some feedback and suggestions for improvement.
Firstly, I appreciate the recognition of the significance of agriculture as a software-intensive sector and the challenges posed by the digital landscape of agriculture. However, I think the background section could benefit from a more in-depth analysis and explanation of the existing solutions for interoperability and data sovereignty in the Agricultural Data Space (ADS). In particular, the Agricultural Knowledge and Innovation Systems (AKIS) should be mentioned and discussed, as they play a crucial role in the development and implementation of digital solutions in agriculture.
Additionally, I noticed that figures 6, 7, and 8 exceed the page limit, which may impact the paper's readability. I would suggest either reducing the size of the figures or finding alternative ways to present the information.
Furthermore, I would like to raise a concern regarding the feasibility of a centralized platform for managing all processes in smart farming, as the agricultural domain is huge and scattered. This is a valid point and should be addressed in the paper. An analysis of the potential benefits and limitations of a centralized platform, as well as alternative solutions and strategies, would strengthen the argument for the reference architecture presented in the paper.
Lastly, I would like to point out some typos, such as line 580, where "going" should be corrected. I believe that careful proofreading and editing of the text would improve the overall quality and clarity of the paper.
Overall, I believe that with these revisions, the paper would better demonstrate the potential and impact of the reference architecture for interoperability and data sovereignty in agriculture, as well as address significant concerns and limitations.
Thank you for considering my feedback. I look forward to seeing the revised version of the paper.
Reviewer 3 Report
The article concerns an important issue of the cooperation of agricultural systems.
The article is structured correctly and the content is presented in a logically consistent order.
In general, I appreciate that the authors refer to Rozanski and Woods when it comes to software architecture. But, the authors focused on the rules of construction of the reference architecture and did not take into account the methods and models of architectural views for designing integration solutions. Interoperability is the main architectural driver in this work so this is a big oversight. The authors should include in the "References" section and comment on in the "Related work" section the following papers: "SOMA: A method for developing service-oriented solutions", (https://doi.org/10.1147/sj.473.0377) and "The 1+5 Architectural Views Model in Designing Blockchain and IT System Integration Solutions", (https://doi.org/10.3390/sym13112000).
Moreover, the aspect of data exchange is not fully addressed in the proposed reference architecture. Again, I suggest referring to the literature on the subject. The following articles show current research results in the field of messaging patterns and data integration: "UML Profile for Messaging Patterns in Service-Oriented Architecture, Microservices, and Internet of Things" (https://doi.org/10.3390/app122412790) and "Data Integration and Interoperability: Towards a Model-Driven and Pattern-Oriented Approach (https://doi.org/10.3390/modelling3010008).
Components in the architecture of the system and cooperating systems should be loosely coupled. However, the authors do not use interfaces. This is a significant disadvantage of the proposed reference architecture.
In addition, the way the architecture is presented also raises objections. For example, Figure 5 should show the Unified Modeling Language (UML) deployment diagram. Figures 6-8 are a kind of combination of content from several UML diagrams. Authors should use separate diagrams: components, communication, and sequence.
As the authors themselves write (lines 641-643), the Data Exchange Manager component has not been implemented or designed. Therefore, the authors did not sufficiently consider the aspect of interoperability. Nevertheless, the authors should enhance that aspect in their approach and the paper.
In the end detailed comment. In Figure 1, the Service Asset Provider should be marked in parentheses as Service Provider and not Consumer.
Round 2
Reviewer 1 Report
The authors have addressed all the concerns I had. Thus, I have no further comments to make.
Author Response
Dear Reviewer 1,
Thank you very much for your feedback.
Rodrigo Falcão (on behalf of all authors).
Reviewer 3 Report
The authors have not addressed all comments.
Once again I am sending all my previously stated comments:
"The article concerns an important issue of the cooperation of agricultural systems.
The article is structured correctly and the content is presented in a logically consistent order.
In general, I appreciate that the authors refer to Rozanski and Woods when it comes to software architecture. But, the authors focused on the rules of construction of the reference architecture and did not take into account the methods and models of architectural views for designing integration solutions. Interoperability is the main architectural driver in this work so this is a big oversight. The authors should include in the "References" section and comment on in the "Related work" section the following papers: "SOMA: A method for developing service-oriented solutions", (https://doi.org/10.1147/sj.473.0377) and "The 1+5 Architectural Views Model in Designing Blockchain and IT System Integration Solutions", (https://doi.org/10.3390/sym13112000).
Moreover, the aspect of data exchange is not fully addressed in the proposed reference architecture. Again, I suggest referring to the literature on the subject. The following articles show current research results in the field of messaging patterns and data integration: "UML Profile for Messaging Patterns in Service-Oriented Architecture, Microservices, and Internet of Things" (https://doi.org/10.3390/app122412790) and "Data Integration and Interoperability: Towards a Model-Driven and Pattern-Oriented Approach (https://doi.org/10.3390/modelling3010008).
Components in the architecture of the system and cooperating systems should be loosely coupled. However, the authors do not use interfaces. This is a significant disadvantage of the proposed reference architecture.
In addition, the way the architecture is presented also raises objections. For example, Figure 5 should show the Unified Modeling Language (UML) deployment diagram. Figures 6-8 are a kind of combination of content from several UML diagrams. Authors should use separate diagrams: components, communication, and sequence.
As the authors themselves write (lines 641-643), the Data Exchange Manager component has not been implemented or designed. Therefore, the authors did not sufficiently consider the aspect of interoperability. Nevertheless, the authors should enhance that aspect in their approach and the paper.
In the end detailed comment. In Figure 1, the Service Asset Provider should be marked in parentheses as Service Provider and not Consumer."
Please bear in mind that a certain level of architectural maturity is required.
The interface is not just a lollipop mark. Besides, I would like the authors to rethink and refine the used UML diagrams.
The manuscript still requires a thorough revision.
Author Response
Dear Reviewer 3,
Thank you for the updated feedback.
The feedback, however, led to further questions on our side, as we could not identify which specific parts needed to be addressed from your point of view. In our responses to your first review, we addressed all of your issues and documented how we addressed them. If the changes made in response to your suggestions were not considered sufficient, or if our explanations were not clear, we would ask you for a direct and constructive response that would help us to address any remaining issues. Thank you very much.
Respectfully,
Rodrigo Falcão (on behalf of all authors).